# Fine Mapping of Five Grain Size QTLs Which Affect Grain Yield and Quality in Rice

**DOI:** 10.3390/ijms25084149

**Published:** 2024-04-09

**Authors:** Yin Zhou, Hanyuan Yang, Enyu Liu, Rongjia Liu, Mufid Alam, Haozhou Gao, Guanjun Gao, Qinglu Zhang, Yanhua Li, Lizhong Xiong, Yuqing He

**Affiliations:** National Key Laboratory of Crop Genetic Improvement and National Center of Plant Gene Research (Wuhan), Hubei Hongshan Laboratory, Huazhong Agricultural University, Wuhan 430070, China; yinzhou@webmail.hzau.edu.cn (Y.Z.); yanghanyuan@webmail.hzau.edu.cn (H.Y.); ley1014@webmail.hzau.edu.cn (E.L.); rongjialiu@mail.hzau.edu.cn (R.L.); mufid.agribhu@gmail.com (M.A.); 2020304120036@webmail.hzau.edu.cn (H.G.); gaoguanjun@mail.hzau.edu.cn (G.G.); qingluzhang@mail.hzau.edu.cn (Q.Z.); liyanhua@mail.hzau.edu.cn (Y.L.); lizhongx@mail.hzau.edu.cn (L.X.)

**Keywords:** grain size, QTL, yield, quality, rice

## Abstract

Grain size is a quantitative trait with a complex genetic mechanism, characterized by the combination of grain length (GL), grain width (GW), length to width ration (LWR), and grain thickness (GT). In this study, we conducted quantitative trait loci (QTL) analysis to investigate the genetic basis of grain size using BC_1_F_2_ and BC_1_F_2:3_ populations derived from two *indica* lines, Guangzhan 63-4S (GZ63-4S) and TGMS29 (core germplasm number W240). A total of twenty-four QTLs for grain size were identified, among which, three QTLs (*qGW1*, *qGW7*, and *qGW12*) controlling GL and two QTLs (*qGW5* and *qGL9*) controlling GW were validated and subsequently fine mapped to regions ranging from 128 kb to 624 kb. Scanning electron microscopic (SEM) analysis and expression analysis revealed that *qGW7* influences cell expansion, while *qGL9* affects cell division. Conversely, *qGW1*, *qGW5*, and *qGW12* promoted both cell division and expansion. Furthermore, negative correlations were observed between grain yield and quality for both *qGW7* and *qGW12*. Nevertheless, *qGW5* exhibited the potential to enhance quality without compromising yield. Importantly, we identified two promising QTLs, *qGW1* and *qGL9*, which simultaneously improved both grain yield and quality. In summary, our results laid the foundation for cloning these five QTLs and provided valuable resources for breeding rice varieties with high yield and superior quality.

## 1. Introduction

Rice (*Oryza sativa* L.) is one of the most important food crops worldwide, providing staple food for more than half of the world’s population [1]. Rice yield is largely determined by three major components: grain weight, number of grains per panicle, and number of effective tillers per plant [2]. Among these, grain weight exhibits a strong correlation with grain size [3]. Grain size is a crucial agronomic trait that has a major impact on the market values of rice grain produce [4,5].

Previous studies have uncovered several signaling pathways controlling grain size, including the guanine nucleotide-binding protein (G protein) signal pathway, the ubiquitin–proteasome pathway, the mitogen-activated protein kinase (MAPK) signaling cascade, the transcriptional regulators pathway, and the phytohormone signaling pathway [3,6,7,8]. G proteins have been demonstrated to regulate grain size in rice. *GS3* is the first identified major QTL that negatively controls grain length and weight [9,10]. The *GS3-3* allele has a longer grain and heavier weight than the *GS3-4* allele [11,12]. GS3 reduces grain length by interacting with RGB1 [13]. Recently, the combination of *OsMADS1^lgy3^*, *dep1*, and *gs3* simultaneously increased both grain yield and quality [14]. Furthermore, the ubiquitin–proteasome pathway could directly or indirectly regulate grain size. *GW2* encodes a RING-type E3 ubiquitin ligase [15] and negatively regulates grain width, weight, and yield [16,17]. *gw2.1*, a new allele of *GW2*, affects grain size by causing cell proliferation, which could be used to improve grain yield and appearance in hybrid breeding [18]. *WTG1* determines grain size by influencing cell expansion [19]. The combination of *npt1* and *dep1-1* has the potential to increase grain yield in rice [20]. In addition, the conserved module of the MAPK signaling cascade plays a key role in the regulation of grain size. The OsMKKK10-OsMKK4-OsMAPK6 cascade modulates grain size by promoting cell division [21,22,23,24,25]. OsWRKY53 can be phosphorylated by OsMAPK6, indicating a cascade of OsMKKK10, OsMKK4, OsMAPK6, and OsWRKY53 in grain size regulation [26]. Moreover, the QTL gene *GLW7* could increase grain length, thickness, and weight by promoting cell expansion in the spikelet hull [27]. Similarly, *GW8* [28,29], *GS2* [30,31], *GS9* [32], and *GL6* [33,34] are all transcriptional regulators that regulate grain size. Additionally, many phytohormones play significant roles in determining grain size. *GW5* was identified as a major QTL for grain width and weight [35]. The decreased expression of *GW5* causes wide and heavy grains, resulting in improved yield [36]. OsPPKL1 is encoded by a major QTL for grain length, *GL3.1* [37,38,39]. *GL3.1* regulates grain size through interaction with OsGSK3 [40]. The *qgl3* allele could increase grain yield without affecting grain quality [39]. The *TGW6* loss-of-function allele enhances grain weight and yield without affecting grain quality [41]. *qTGW3* negatively regulates grain size and weight [42,43,44]. A loss-of-function mutation in *qTGW3* results in large and heavy grains, suggesting this locus has potential in breeding high-yield rice [44].

So far, numerous grain size QTLs have been identified, among which a few major QTLs have been cloned. However, the genetic basis of grain size is still not well elucidated, and mining novel QTLs for grain size is of great importance to gain a better understanding of regulation mechanisms and provide gene resources for breeding applications. In this study, we mapped QTLs for grain size using BC_1_F_2_ and BC_1_F_2:3_ populations derived from the cross between GZ63-4S and W240. A total of twenty-four QTLs were identified, among which five major QTLs were confirmed and further fine mapped. Scanning electron microscopic analysis and expression analysis revealed the cytological basis underlying the five major QTLs on grain size. What is more, an investigation of yield- and quality-related traits demonstrated the potential of these QTLs for breeding rice with high yield and superior quality.

## 2. Results

### 2.1. Phenotypic Variation and Correlation in the BC_1_F_2_ and BC_1_F_2:3_ Populations

The BC_1_F_2_ and BC_1_F_2:3_ populations, as well as the two parents, exhibited great phenotypic variation in grain size. Compared with GZ63-4S, W240 exhibited greater grain size (Appendix A). In two years of repeated field trials, GL ranged from 8.64 mm to 10.27 mm in the BC_1_F_2_ population and from 8.58 mm to 10.23 mm in the BC_1_F_2:3_ population, respectively (Figure 1A). The ranges of GW were 2.37–2.98 mm and 2.43–2.92 mm (Figure 1B), LWR’s were 2.96–3.92 and 3.11–4.04 (Figure 1C), and GT’s were 1.92–2.19 mm and 1.94–2.22 mm (Figure 1D). Notably, all these traits exhibited normal distribution in both years, indicating typical patterns of quantitative variation. The results suggested that both BC_1_F_2_ and BC_1_F_2:3_ populations met the requirement for QTL mapping.

The correlation coefficients among the four grain traits in two years showed that both GL and GW exhibited weak to moderate correlations with LWR and GT in 2016 or 2017. LWR displayed no and weak correlation with GT in 2016 and 2017, respectively (Appendix A). Moreover, GL16 and GL17, GW16 and GW17, LWR16 and LWR17, and GT16 and GT17 all displayed moderate correlation (Appendix A).

### 2.2. QTL Mapping and Verification

To identify QTLs associated with GL, GW, GT, and TGW, 177 polymorphic markers were used to screen 327 plants from the BC_1_F_2_ population. We generated a BC_1_F_2:3_ population by propagating single plants from the BC_1_F_2_ populations for an additional generation. The phenotypic evaluation was conducted using mixed single plants from each family in the BC_1_F_2:3_ population, with the average phenotype serving as the phenotype for each individual. Subsequent QTL analysis revealed the presence of twenty-four QTLs, including eleven for GL, five for GW, six for LWR, and two for GT (Figure 2, Table 1).

Based on the above QTL mapping, *qGW1*, *qGW5*, *qGW7*, *qGL9*, and *qGW12* were identified as significant contributors to grain size (Table 1). Five BC_1_F_6_ and BC_1_F_7_ populations developed from the BC_1_F_2:3_ line that were heterozygous in the target QTL region and homozygous for most of the other grain size QTLs were chosen to precisely evaluate their effects on grain size, respectively.

### 2.3. The Genetic Effect of qGW1

The grain width showed significant differences between NIL with homologous W240 (NIL-W) and NIL with homologous GZ63-4S (NIL-G) for *qGW1* (Figure 3A,B and Appendix A). To fine map *qGW1*, we developed a BC_1_F_8_ population consisting of 2994 plants and subsequently identified 478 recombinants using markers RM128 and RM319. Eventually, we localized *qGW1* to a high-resolution linkage map by progeny testing 36 recombinants and narrowed the *qGW1* locus to a 231 kb region between markers R1344 and R1346 (Figure 3C).

Grain size is restricted by the spikelet hull, which is determined by cell division and expansion [45]. We conducted scanning electron microscopic (SEM) analysis to uncover the cytological basis underlying the regulation of grain size in *qGW1*. The values of cell width and the number of transverse cells were higher in NIL-W*^qGW1^* than in NIL-G*^qGW1^* (Figure 3D–I). Furthermore, we examined the expression of cell cycle and cell expansion genes in young panicles of NILs using qRT-PCR. Our findings indicated that four cell cycle related-genes (*E2F2*, *CDKA1*, *CYCA3;1*, and *CYCT1*), as well as four cell expansion related-genes (*EXPA10*, *EXPA5*, *EXPA6*, and *EXPA7*), were up-regulated in NIL-W*^qGW1^*, suggesting that *qGW1* regulates grain width by promoting both cell division and cell expansion (Figure 3J and Figure 9).

### 2.4. The Genetic Effect of qGW5

Compared with NIL-W*^qGW5^*, the values of grain length were significantly higher in NIL-G*^qGW5^* (Figure 4A,B and Appendix A). A progeny test of homozygous segregates further narrowed down the *qGW5* locus to a 128 kb region between R5101 and R51142 (Figure 4C).

NIL-G*^qGW5^* displayed significantly larger cell length and a higher number of longitudinal cells than that of NIL-W*^qGW5^* (Figure 4D–I). Expression analysis revealed that five genes related to the cell cycle (*E2F2*, *CDKA2*, *CYCA3;1*, *MAPK*, and *MCM3*) and four genes related to the cell expansion (*EXPA10*, *EXPA3*, *EXPA4*, and *EXPB7*) showed higher expression levels in NIL-G*^qGW5^*. These findings revealed that *qGW5* regulates grain length by influencing both cell division and expansion in the spikelet hull (Figure 4J and Figure 9).

### 2.5. The Genetic Effect of qGW7

The significant variations in grain width were observed among lines carrying different genotypes of *qGW7* (Figure 5A,B and Appendix A). The *qGW7* locus was ultimately mapped to a region of 444 kb between R7277 and R7281 using 1680 plants from the BC_1_F_9_ population (Figure 5C).

The NIL-W*^qGW7^* spikelet hull had a wider cell size than NIL-G*^qGW7^*, while no difference was observed in cell number (Figure 5D–I). We further observed an up-regulation in the expression of six cell expansion related-genes (*EXPA10*, *EXPA3*, *EXPA5*, *EXPA6*, *EXPA7* and *EXPB7*) in NIL-W*^qGW7^*, indicating that *qGW7* regulates grain width through the alteration of cell expansion (Figure 5J and Figure 9).

### 2.6. The Genetic Effect of qGL9

NIL-G*^qGL9^* had significantly larger grain length than NIL-W*^qGL9^* (Figure 6A,B and Appendix A). Subsequently, we mapped the *qGL9* to a 335 kb region between markers R9194 and R9197 in the BC_1_F_9_ generation, which included a total of 1584 plants (Figure 6C).

The number of longitudinal cells in the spikelet hull was higher in NIL-G*^qGL9^* than in NIL-W*^qGL9^*. However, there was no difference in cell length between these two NILs (Figure 6D–I). Notably, NIL-G*^qGL9^* exhibited up-regulated expression levels of fifteen genes related to cell cycle (*CDC20*, *CDKA2*, *CDKB*, *CYCA2.1*, *CYCA3;2*, *CYCB1;1*, *CYCB2.2*, *CYCD1;1*, *CYCD4*, *CYCD6*, *H1*, *KN*, *MAPK*, *MCM2*, and *MCM4*), resulting in increased cell division within the spikelet and ultimately leading to an increase in grain length (Figure 6J and Figure 9).

### 2.7. The Genetic Effect of qGW12

We constructed NIL-G*^qGW12^* and confirmed that this allele could significantly increase grain width (Figure 7A,B and Appendix A). Subsequently, *qGW12* was mapped within a 624 kb interval flanked by markers R12246 and R12252 (Figure 7C).

Analysis of the outer glume found that NIL-G*^qGW12^* exhibited increased cell number and larger cell size in the grain-width direction, causing wider grain (Figure 7D–I). In addition, the expression levels of six genes related to cell cycle (*CDKA1*, *CYCB1;1*, *CYCD1;1*, *CYCD3*, *CYCIaZm*, and *KN*) and three genes related to cell expansion (*EXPA6*, *EXPB4*, and *EXPB7*) were higher in NIL-G*^qGW12^* than in NIL-W*^qGW12^*. Thus, the up-regulation of both cell division and expansion genes was responsible for the increase in the grain width of NIL-G*^qGW12^* (Figure 7J and Figure 9).

### 2.8. Investigation of Traits Related to Rice Yield and Quality in Five NILs

Finally, we performed phenotypic comparisons of yield- and quality-related traits among five NILs. The grain length of NIL-G*^qGW1^*, NIL-W*^qGW5^*, and NIL-W*^qGL9^* exhibited significant reductions than their respective NILs, while no differences were observed in other NILs. Phenotypic variations were detected among the five NILs for grain width and length to width ration. Specifically, larger variations in 1000-grain weight and the number of tillers per plant were found in NIL-W*^qGW1^* and NIL-W*^qGL9^*, whereas higher plant height and the number of filled grains per panicle were observed in NIL-G*^qGW7^* and NIL-W*^qGW12^*. Grain yield per plant was significantly higher in NIL-G*^qGW1^*, NIL-G*^qGW7^*, NIL-G*^qGL9^*, and NIL-W*^qGW12^* than that of their respective NILs. However, there were no differences in grain yield per plant between NIL-W*^qGW5^* and NIL-G*^qGW5^* (Figure 8A–H). These results suggested that the increased yield of *qGW1* and *qGL9* were primarily attributed to enhanced number of tillers per plant, while the increased yield of *qGW7* and *qGW12* were mainly due to enhanced numbers of filled grains per panicle.

There were no differences in albumin content among the five NILs. In contrast, three, four and five NILs exhibited remarkable variations in globulin, prolamin, and glutenin content, respectively. Compared with NIL-W*^qGW5^*, NIL-G*^qGW5^* displayed higher total starch and amylose content, whereas no differences were observed in the other four NILs. NIL-W*^qGW5^* and NIL-G*^qGL9^* exhibited significant increases in gel consistency, while NIL-G*^qGW1^*, NIL-G*^qGW5^*, NIL-W*^qGW7^*, NIL-G*^qGL9^*, and NIL-G*^qGW12^* showed huge improvements in taste score (Figure 8I–P). These findings indicated that the improved cooking and eating quality of *qGW1* and *qGL9* results from reduced globulin and glutenin content, whereas the improved quality of *qGW7* and *qGW12* derives from decreased prolamin and glutenin content. In addition, both decreased globulin and glutenin content and increased total starch content, amylose content, and gel consistency lead to an increase in the cooking and eating quality of *qGW5*. Usually, rice with lower protein content but higher amylose content and gel consistency exhibits superior cooking and eating quality [46].

In summary, negative correlations were observed between yield and rice quality for both *qGW7* and *qGW12*. The impact of *qGW5* on yield was negligible. Conversely, it exerted significant influences on quality-related traits, suggesting that this locus could enhance rice quality without compromising yield. Meanwhile, we provide two promising QTLs, *qGW1* and *qGL9*, for breeding rice with a high yield and superior quality.

## 3. Discussion

### 3.1. QTL Mapping for Grain Size and Weight

In previous studies of QTL localization, mapping populations were typically constructed using parents with significant phenotypic and genetic disparities, resulting in the gradual cloning of dominant QTLs, such as *GS3* [9,10], *GS2* [30,31], *GW2* [15], and *GW5* [35,36].

As a primary isolated mapping population, the BC_1_F_2_ population offers the advantages of being relatively simple to construct, requiring less time, and providing rich genetic information. We generated a BC_1_F_2:3_ population by propagating single plants from the BC_1_F_2_ populations for an additional generation to address issues of offspring segregation and reproductive limitations caused by the BC_1_F_2_ population. Until now, many QTLs or genes have been reported from F_2_ and F_2:3_ populations, such as *Rpsan 1*, *qPH9*, *FM1*, *Co-1HY*, *YrZ15-1949*, *qHBV4.2*, *qHBV6.1*, *qHBV11.1*, and *qHBV11.2* [47,48,49,50,51,52]. To identify novel QTLs controlling grain size, we developed BC_1_F_2_ and BC_1_F_2:3_ populations derived from W240 and GZ63-4S. A total of twenty-four QTLs for grain size were detected in this study (Figure 2, Table 1). By comparing chromosome positions and molecular markers, it was found that *OsCKX1* exerts a negative regulatory effect on grain size within the same region as *qGL1* [53]. *GW2*, which negatively controls grain width, was identified at the *qGL2.2* locus [15]. Zhao et al. [54] detected two QTLs (*qGL2.1* and *qGW4*) in regions similar to those of *qGL2.1* and *qGW4*. Additionally, two other QTLs for grain length (*qGL8.1* and *qGL5*) were, respectively, located in the vicinity of QTLs detected in previous studies [55,56].

Notably, we discovered five novel QTLs, *qGW1*, *qGW5*, *qGW7*, *qGL9*, and *qGW12*, which made significant contributions to grain size. Taken together, these results proved the reliability of using BC_1_F_2_ and BC_1_F_2:3_ populations for rice QTL mapping.

### 3.2. Validation and Fine Mapping of the Five QTLs

In previous studies, many genes controlling grain size have been cloned using the map-based cloning approach, such as *GW2* [15], *GS2* [30,31], and *GL3.1* [37,38,39]. Therefore, QTL mapping, validation, and fine mapping are essential steps in the process of gene cloning.

In this study, we successfully validated five QTLs (*qGW1*, *qGW5*, *qGW7*, *qGL9*, and *qGW12*) that exert significant influences on grain size and narrow down their locations to regions ranging from 128 kb to 624 kb using a map-based cloning method. These results laid the foundation for the additional fine mapping of the five QTLs, the cloning of the candidate genes, and functional research to explore the genetic mechanisms underlying grain size.

### 3.3. Candidate Gene Analysis of the Five QTLs

According to the Nipponbare genome, *LOC_Os01g59660* (*MYBGA*), *LOC_Os01g59760* (*OsbZIP09*), and *LOC_Os01g59780* are potential candidate genes for *qGW1*. Among these genes, *LOC_Os01g59660*, which encodes an MYB family transcription factor, has been shown to influence the development of floral organs, tiller number, and grain yield [57,58]. As we all know, MYB proteins play critical roles in development, metabolism, biotic stresses, and abiotic stresses [59,60]. Other members of the MYB family transcription factor, *LOC_Os01g49160* (*RGN1*) and *LOC_Os08g06110* (*Nhd1*), have been reported to affect rice gain size [61,62], while *LOC_Os01g7402* (*OsLUX*) has been displayed to influence rice yield and heading date [63]. What is more, mutants of *LOC_Os01g59760* (*OsbZIP09*), encoding a bZIP transcription factor, exhibited longer seeds compared with the wild type [64]. It has been reported that *LOC_Os09g34880* (*OsbZIP76*), *LOC_Os06g50600* (*OsbZIP55*), and *LOC_Os06g15480* (*OsbZIP47*) function as regulators of grain size [65,66,67]. Furthermore, *LOC_Os01g59780* encodes a protein that contains an AP2 domain. The homologous family genes, including *LOC_Os07g47330* (*FZP*), *LOC_Os05g03040* (*RSR1*), *LOC_Os05g27930* (*OsDREB2B*), and *LOC_Os07g13170* (*OsSNB*), have been studied for their role in controlling seed size in rice [68,69,70,71,72].

In the *qGW5* interval, *LOC_Os05g03020*, a C2H2 zinc finger protein, could potentially be considered as a candidate gene. *LOC_Os04g36650* (*NSG1*) and *LOC_Os06g48530* (*Du13*), belonging to the C2H2 zinc finger family, have been found to determine grain size [73,74].

*LOC_Os07g46460* (*spl32*) and *LOC_Os07g46590* may be the candidate genes for *qGW7*. Compared with the wide type, the *spl32* mutant exhibited a decrease in grain size [75]. Moreover, *LOC_Os03g51230* (*OsDDM1b*), a homologous member of the gene family that contains *LOC_Os07g46590*, has been reported to regulate grain size [76].

There are three putative genes in the qGL9 region, which include *LOC_Os09g32944* (*OsSPL18*), *LOC_Os09g32740* (*OsLMP1*), and *LOC_Os09g32948* (*OsMADS8*). Compared with the wild type, the mutant of *OsSPL18* exhibited narrower and thinner grains [77]. Other members of the SPL family, such as *OsSPL13*, *OsSPL12*, and *OsSPL16*, played crucial roles in determining grain size [27,29,78]. Furthermore, *LOC_Os02g14730* (*LG1*), a homologous member of the gene family containing *LOC_Os09g32740*, has been reported to positively control grain size [79]. What is more, MADS proteins play crucial roles in flower and fruit development [80]. Previous studies have demonstrated that *OsMADS1*, *OsMADS34*, and *OsMADS56* exert significant influence on grain size [14,81,82].

*LOC_Os12g40190* (*OsXLG4*), *LOC_Os12g40570* (*OsWRKY83*), *LOC_Os12g40460*, and *LOC_Os12g40830* are potential candidate genes for *qGW12*. Firstly, *RGA1*, a homologous member of the gene family containing *LOC_Os12g40190* (*OsXLG4*), positively regulates rice grain size [83]. Secondly, *OsWRKY36*, a homologous member of the gene family containing *LOC_Os12g40570* (*OsWRKY83*), has been reported to suppress grain size by inhibiting GA signaling [84]. Thirdly, *DGS1*, a homologous member of the gene family containing *LOC_Os12g40460*, played a positive role in regulating grain size by binding to OsBZR1 [85]. The mutant of *FRRP1* resulted in an increase in rice grain length [86]. Finally, *OsFRK3*, a homologous member of the pfkB family containing *LOC_Os12g40830*, was identified as a positive regulator of grain width and thickness through its influence on sugar metabolism [87]. In the future, transgenic studies will be conducted on these five QTLs to further elucidate the molecular mechanisms underlying grain size.

### 3.4. Cytological Analysis of the Five QTLs

To elucidate the characterization and commonality of differentially expressed genes influencing cell number and size in *qGW1*, *qGW5*, *qGW7*, *qGL9*, and *qGW12*, we analyzed the association among these genes involved in cell division and expansion. Specifically, we identified one specific differential gene in *qGW1*, two in *qGW5*, ten in *qGL9*, and three in *qGW12*. The regulatory modules of *qGW1*-*qGW5*, *qGW1*-*qGW12*, *qGW5*-*qGL9*, and *qGL9*-*qGW12* exhibited differential expression in cell cycle-related genes, including *E2F2* and *CYCA3;1*; *CDKA1*; *MAPK* and *CDKA2*; and *CYCD1;1*, *CYCB1;1*, and *KN*, respectively (Figure 9A,B). These findings suggest that the regulation of grain width by *qGW1* and *qGW12* is mediated through the modulation of *CDKA1*, while the influence of *qGW5* and *qGL9* on grain length involves *MAPK*- and *CDKA2*-mediated mechanisms, thereby altering cell numbers. Moreover, the differential expression of cell expansion-related genes, such as *EXPA5*, *EXPA10*, *EXPA6* and *EXPA7*; *EXPA3*, *EXPA10*, and *EXPB7*; *EXPA10*; and *EXPA6* and *EXPB7*, were observed within the regulatory modules of *qGW1*-*qGW7*, *qGW5*-*qGW7*, *qGW1*-*qGW5*-*qGW7*, *qGW1*-*qGW7*-*qGW12*, and *qGW5*-*qGW7*-*qGW12*, respectively (Figure 9A,B), suggesting *qGW1*-*qGW7*-*qGW12* influences cell size and grain width by altering the expression of *EXPA6*. Numerous QTLs and genes have been reported to regulate grain size by influencing the expression of cell division and expansion-related genes, such as *qTGW2b* [88], *GLW7.1* [89], *GL3.1* [38], and *GS2* [90]. In addition, negative correlations were observed between gene expression and grain yield for both *qGW1* and *qGW7*. Despite variations in the expression of cell cycle- and expansion-related genes, there was no alteration in the yield of *qGW5*. Notably, positive correlations were observed between gene expression and grain yield for both *qGL9* and *qGW12*. Variations in the expression of genes associated with cell division and expansion not only impact grain size, but also influence grain yield [38,89,90].

Understanding the relationship between different genes involved in cell division and expansion is convenient for studying the shared characteristics and cytological molecular mechanisms underlying the influence of these five QTLs on rice grain size.

### 3.5. Potential Uses of the Five QTLs in Rice Breeding

High yield and superior quality are essential goals of rice breeders. Gene cloning and molecular breeding have become important techniques to breed high-yield and superior-quality varieties. The major genes that increase grain size, such as *GW2* [15], *GS2* [31], and *GW5* [35,36], produce higher grain yield while simultaneously reducing grain quality. The *gs9* allele has the potential to improve the appearance quality of milled rice without affecting grain yield [32]. Nevertheless, research advances have revealed several genes that could be utilized to help breeders develop new elite rice varieties with high yield and superior quality. The combination of the *OsMADS1^lgy3^* allele with *dep1-1* and *gs3* alleles has the potential to simultaneously improve both grain yield and quality in rice [14]. *GLW7.1* also represents a novel way to breed high-yield and superior-quality varieties [89].

In our study, negative correlations were observed between yield and rice quality for both *qGW7* and *qGW12*. However, *qGW5* exhibited the potential to enhance quality without compromising yield. Interestingly, *qGW1* and *qGL9* displayed positive correlations between grain yield and rice quality, indicating their pleiotropic effects in simultaneously improving both yield and quality. In summary, the identification of genetic resources, such as *qGW5*, *qGW1*, and *qGL9*, provides a theoretical foundation for breeding strategies aimed at enhancing grain yield and quality in rice. Our results laid the foundation for cloning these five genes. Additionally, such information will help breeders to improve grain yield and quality in rice.

## 4. Materials and Methods

### 4.1. Population Development and Field Experiment

The BC_1_F_2_ and BC_1_F_2:3_ populations were derived from two *indica* lines, GZ63-4S (the recurrent parent) and W240 (the donor parent). GZ63-4S is a photoperiod-thermo-sensitive genic male sterile *indica* rice line, carrying the infertility gene of *TMS5* [91]. W240 is an *indica* variety with larger grain size. The BC_1_F_2_ and its derived BC_1_F_2:3_ populations were used for QTL mapping. To validate the genetic effects of *qGW1*, *qGW5*, *qGW7*, *qGL9*, and *qGW12*, five BC_1_F_6_ and BC_1_F_7_ populations were planted at the experimental station of Huazhong Agricultural University at Wuhan, Hubei province and Lingshui, Hainan province in 2020, respectively. The progeny tests were conducted in the BC_1_F_8_ and BC_1_F_9_ generations. The BC_1_F_2_, BC_1_F_2:3_, BC_1_F_6_, BC_1_F_7_, BC_1_F_8_, and BC_1_F_9_ populations were planted in 2016, 2017, 2020 (twice), and 2021 (twice). The detailed process of population development is shown in Appendix A. All rice plants with a density of 16 cm × 26 cm were grown under normal field management. Field management followed local practices. Ten plants were harvested from the middle of each row for trait measurement.

### 4.2. Trait Measurement

Harvested rice grains from each plant were air-dried and stored at room temperature for three months before testing. Grain length, grain width, grain number, grain yield, and 1000-grain weight were measured using the yield traits scorer (YTS) platform [92], whereas grain thickness was measured using vernier calipers. The plant height was measured from the main culm. The number of tillers per plant was counted as all fertile panicles in one plant. Additionally, flour ground from milled grain was used to determine the albumin content, globulin content, prolamin content, glutenin content, total starch content, amylose content, and gel consistency according to the NY/T 593-2013 standard published by the Ministry of Agriculture, China (http://www.zbgb.org/27/StandardDetail1476335.htm, accessed on 3 October 2019). Taste scores for milled rice were evaluated using a taste analyzer kit (Satake, RLTA10B-KC, Hiroshima, Japan) [93].

### 4.3. Genetic Map Construction and QTL Mapping

The parent varieties GZ63-4S and W240 were sequenced using the illumine HiSeq2000 (Illumina, San Diego, CA, USA), and the sequencing data were compared and assembled according to the rice reference genome (Rice Genome Annotation Project, http://rice.uga.edu/, accessed on 6 February 2022) [94]. All mapping primers were designed in reference to the sequencing data of two parents. A total of 67 polymorphic simple sequence repeat (SSR) markers, 103 insert and deletion (InDel) markers, and 7 Kompetitive allele-specific PCR (KASP) markers were evenly distributed across 12 chromosomes to genotype the 327 BC_1_F_2_ lines. According to the cetyltrimethylammonium bromide (CTAB) method, genomic DNA was extracted from leaves [95]. The genotyping was carried out using 4% Polyacrylamide gels (PAGE) migration, as previously reported by Panaud et al. [96]. DNA bands on PAGE gel were displayed by silver nitrate staining and NaOH-formaldehyde solution. Combining the genotype data from the BC_1_F_6_ lines and the phenotype data from both BC_1_F_6_ and BC_1_F_7_ lines, we employed the Kosambi mapping function of MapMaker/Exp3.0 program to construct a genetic linkage map [97]. QTL analysis was performed using the composite interval mapping method with Windows QTL cartographer 2.5 software (WinQTLCart 2.5) [98].

Refraining from considering the QTL of cloned genes, we selected five major QTLs that have higher LOD, Add, and PVE to study. The genotypes of the five BC_1_F_6_ and BC_1_F_7_ lines of *qGW1*, *qGW5*, *qGW7*, *qGL9*, and *qGW12* were determined using two flanking markers within the mapping interval of QTLs. To fine map these QTLs, we developed five BC_1_F_9_ populations consisting of 1728, 2046, 1680, 1584, and 1440 individuals, respectively. Another 6, 5, 14, 5, and 6 specific markers were developed to genotype the recombinants of these QTLs. Relevant primer sequences are shown in Appendix A.

### 4.4. Scanning Election Microscopy

Lemmas of spikelets at the heading stage were collected for scanning electron microscopy, fixed in FAA solution (50% ethanol, 5% glacial acetic acid, and 3.7% formaldehyde) at 4 °C for 24 h. The young panicles were sampled at the length of about 3 cm. Then the samples were coated with gold under vacuum conditions, and observed using a scanning electron microscope (JEOL, JSM-6390LV, Tokyo, Japan) under 10 kV acceleration voltage and a 30 nm spot size. Cell number and cell size were calculated at 50 × and 100 × magnification, respectively. The spikelet epidermal cell size was measured using Image J software (NIH), and cell number was counted manually. Scanning electron microscopy analysis involved at least three biological replications of mounted specimens.

### 4.5. RNA Extraction, Reverse Transcription, and qRT-PCR

Total RNA was extracted from young panicles using the TRIzol method (Invitrogen, 15596026, Shanghai, China), and then treated with RNase-free DNase I (Invitrogen, 15596026, Shanghai, China). First, strand cDNA was reverse-transcribed using the M-MLV Reverse Transcriptase kit (Promega, M170A, Madison, WI, USA). All procedures were carried out according to the manufacturer’s protocol. qRT-PCR was performed using ABI Real-Time PCR system with the SYBR Green I mix (TaKaRa, Shiga, Japan) according to the manufacturer’s instructions. *OsActin* gene was used as an internal control to normalize gene expression. The gene expression levels in three biological replicates and three technical replicates were calculated to evaluate the significance of differences between samples using the student’s *t*-test. Relevant primer sequences are given in Appendix A.

### 4.6. Statistical Analysis

Differences between two sets of data were presented as the mean ± standard deviation and performed using the student’s *t*-test. We conducted differential expression analysis of differentially expressed genes using Cytoscape software (3.9.1) [99] and the Metware Cloud, a free online platform for data analysis (https://cloud.metware.cn, accessed on 1 January 2024).

## Figures and Tables

**Figure 1 ijms-25-04149-f001:**
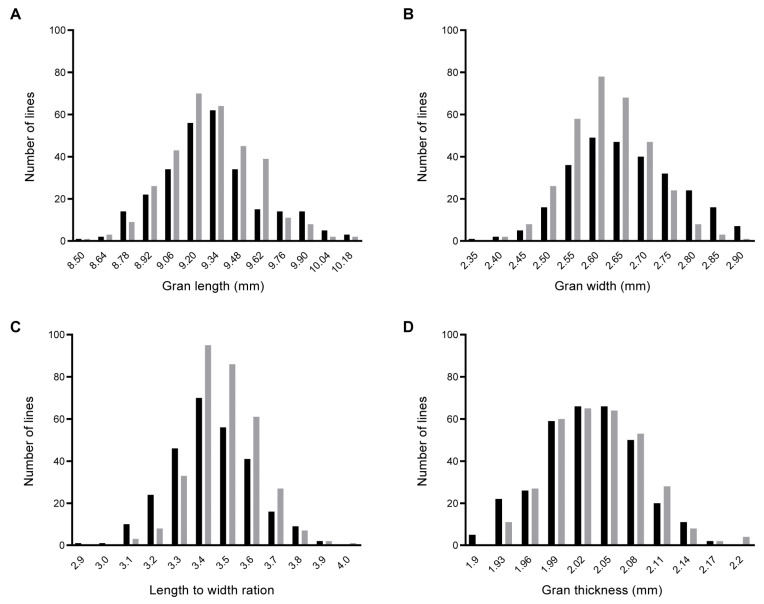
Frequency distributions of GL (**A**), GW (**B**), LWR (**C**), and GT (**D**) in the BC_1_F_2_ and BC_1_F_2:3_ populations. The vertical axis represents the number of BC_1_F_2_ and BC_1_F_2:3_ plants, with black and gray bars, respectively.

**Figure 2 ijms-25-04149-f002:**
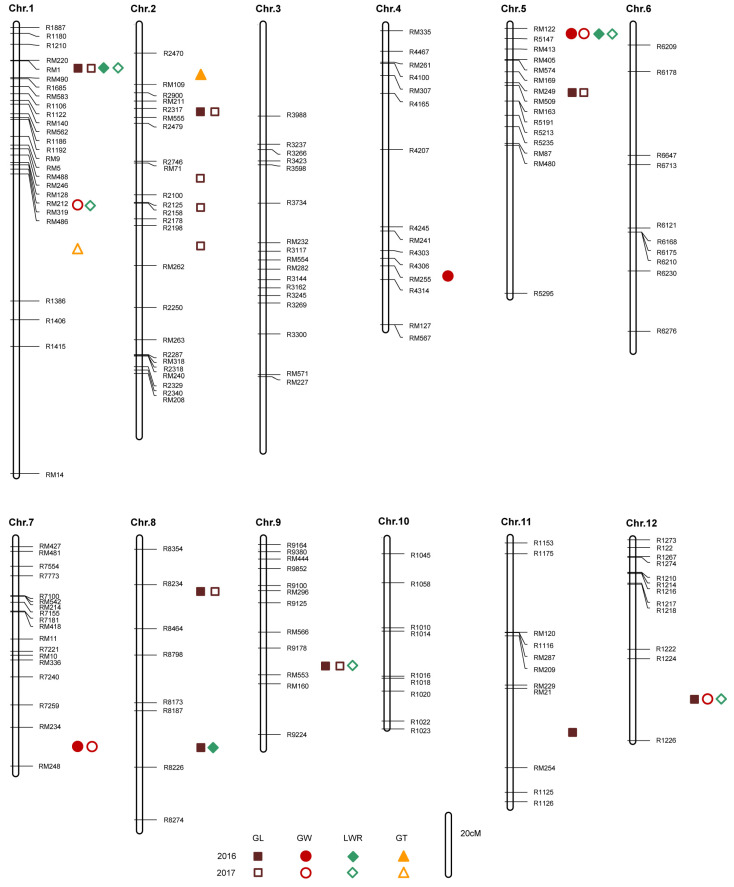
Genetic linkage map of grain-size-related QTLs detected in the BC_1_F_2_ and BC_1_F_2:3_ populations.

**Figure 3 ijms-25-04149-f003:**
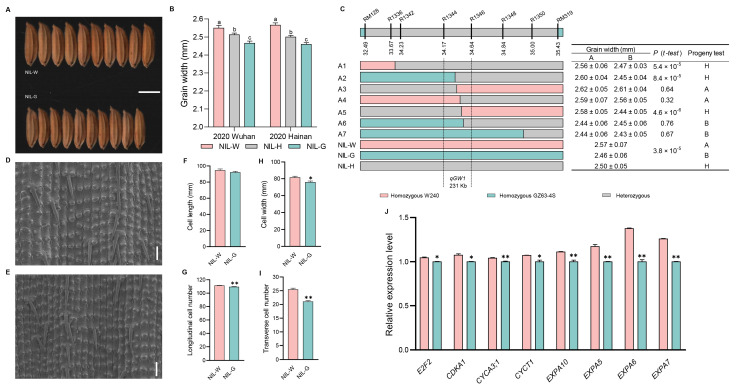
Analysis of *qGW1* influence grain width. (**A**) Grain morphology. Scale bar: 5 mm. (**B**) Grain width difference among three haplotypes in 2020. (**C**) Fine mapping of *qGW1*. The numbers below the bar are physical distance (Mb). (**D**,**E**) Scanning electron microscopy of the outer epidermal cells of NIL-W*^qGW1^* and NIL-G*^qGW1^*. Scale bar: 100 µm. (**F**) Cell length. (**G**) Total number of longitudinal cells. (**H**) Cell width. (**I**) Total number of transverse cells. (*n* = 10). (**J**) qRT-PCR analysis of four cell cycle related-genes and four cell expansion related-genes between NILs of *qGW1*. Data are represented as mean ± s.e.m. (*n* = 9). Duncan’s multiple range tests were used to conduct statistical analysis (a, b and c indicate *p* < 0.01). The student’s *t*-test was used to produce *p* values (*, ** indicate significance at *p* < 0.05 and *p* < 0.01, respectively).

**Figure 4 ijms-25-04149-f004:**
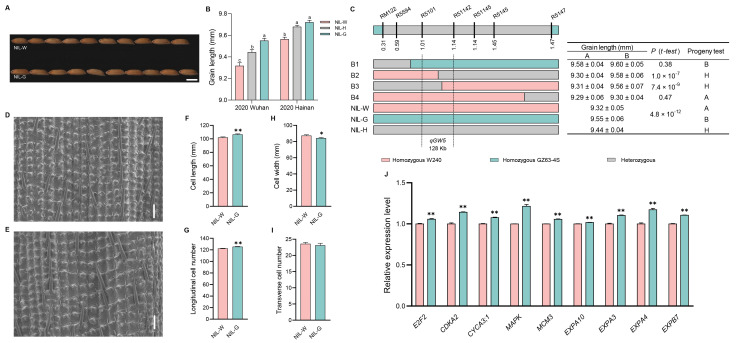
Analysis of *qGW5* influence grain length. (**A**) Grain morphology. Scale bar: 5 mm. (**B**) Grain length difference among three haplotypes in 2020. (**C**) Fine mapping of *qGW5*. The numbers below the bar are physical distance (Mb). (**D**,**E**) Scanning electron microscopy of the outer epidermal cells of NIL-W*^qGW5^* and NIL-G*^qGW5^*. Scale bar: 100 µm. (**F**) Cell length. (**G**) Total number of longitudinal cells. (**H**) Cell width. (**I**) Total number of transverse cells. (*n* = 10). (**J**) qRT-PCR analysis of five cell cycle related-genes and four cell expansion related-genes between NILs of *qGW5*. Data are represented as mean ± s.e.m. (*n* = 9). Duncan’s multiple range tests were used to conduct statistical analysis (a, b and c indicate *p* < 0.01). The student’s *t*-test was used to produce *p* values (*, ** indicate significance at *p* < 0.05 and *p* < 0.01, respectively).

**Figure 5 ijms-25-04149-f005:**
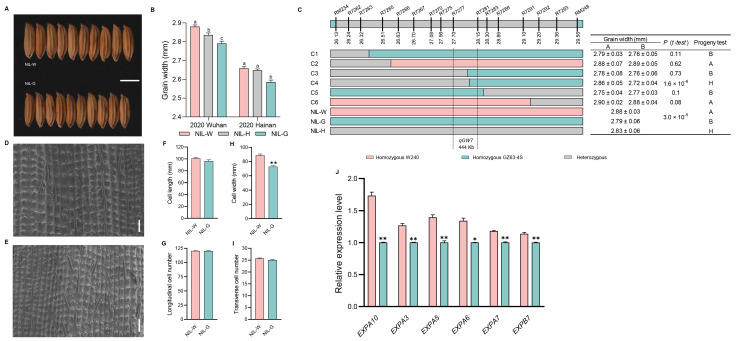
Analysis of *qGW7* influence grain width. (**A**) Grain morphology. Scale bar: 5 mm. (**B**) Grain width difference among three haplotypes in 2020. (**C**) Fine mapping of *qGW7*. The numbers below the bar are physical distance (Mb). (**D**,**E**) Scanning electron microscopy of the outer epidermal cells of NIL-W*^qGW7^* and NIL-G*^qGW7^*. Scale bar: 100 µm. (**F**) Cell length. (**G**) Total number of longitudinal cells. (**H**) Cell width. (**I**) Total number of transverse cells. (*n* = 10). (**J**) qRT-PCR analysis of six cell expansion related-genes between NILs of *qGW7*. Data are represented as mean ± s.e.m. (*n* = 9). Duncan’s multiple range tests were used to conduct statistical analysis (a, b and c indicate *p* < 0.01). The student’s *t*-test was used to produce *p* values (*, ** indicate significance at *p* < 0.05 and *p* < 0.01, respectively).

**Figure 6 ijms-25-04149-f006:**
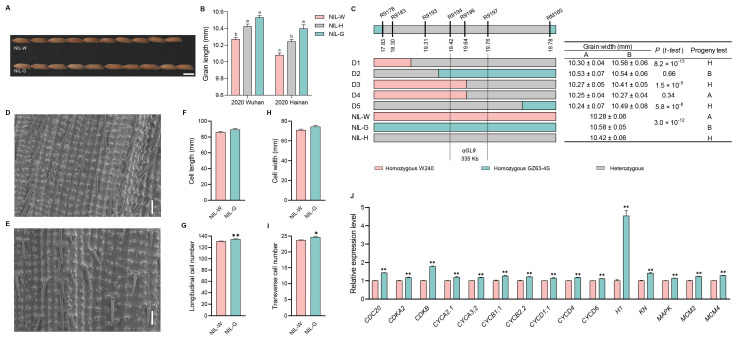
Analysis of *qGL9* influence grain length. (**A**) Grain morphology. Scale bar: 5 mm. (**B**) Grain length difference among three haplotypes in 2020. (**C**) Fine mapping of *qGL9*. The numbers below the bar are physical distance (Mb). (**D**,**E**) Scanning electron microscopy of the outer epidermal cells of NIL-W*^qGL9^* and NIL-G*^qGL9^*. Scale bar: 100 µm. (**F**) Cell length. (**G**) Total number of longitudinal cells. (**H**) Cell width. (**I**) Total number of transverse cells. (*n* = 10). (**J**) qRT-PCR analysis of fifteen cell cycle related-genes between NILs of *qGL9*. Data are represented as mean ± s.e.m. (*n* = 9). Duncan’s multiple range tests were used to conduct statistical analysis (a, b and c indicate *p* < 0.01). The student’s *t*-test was used to produce *p* values (*, ** indicate significance at *p* < 0.05 and *p* < 0.01, respectively).

**Figure 7 ijms-25-04149-f007:**
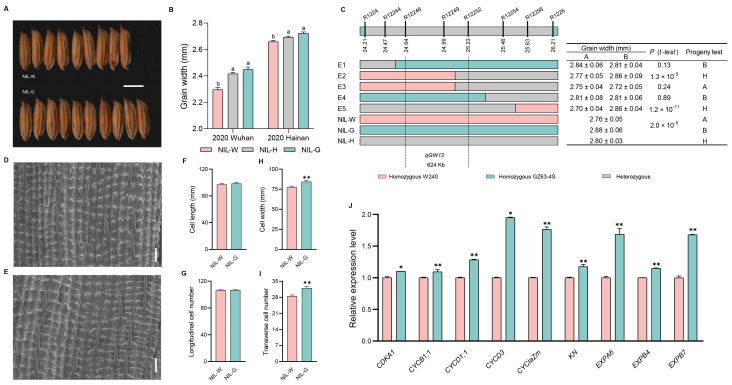
Analysis of *qGW12* influence grain width. (**A**) Grain morphology. Scale bar: 5 mm. (**B**) Grain width difference among three haplotypes in 2020. (**C**) Fine mapping of *qGW12*. The numbers below the bar are physical distance (Mb). (**D**,**E**) Scanning electron microscopy of the outer epidermal cells of NIL-W*^qGW12^* and NIL-G*^qGW12^*. Scale bar: 100 µm. (**F**) Cell length. (**G**) Total number of longitudinal cells. (**H**) Cell width. (**I**) Total number of transverse cells. (*n* = 10). (**J**) qRT-PCR analysis of six cell cycle related-genes and three cell expansion related-genes between NILs of *qGW12*. Data are represented as mean ± s.e.m. (*n* = 9). Duncan’s multiple range tests were used to conduct statistical analysis (a and b indicate *p* < 0.01). The student’s *t*-test was used to produce *p* values (*, ** indicate significance at *p* < 0.05 and *p* < 0.01, respectively).

**Figure 8 ijms-25-04149-f008:**
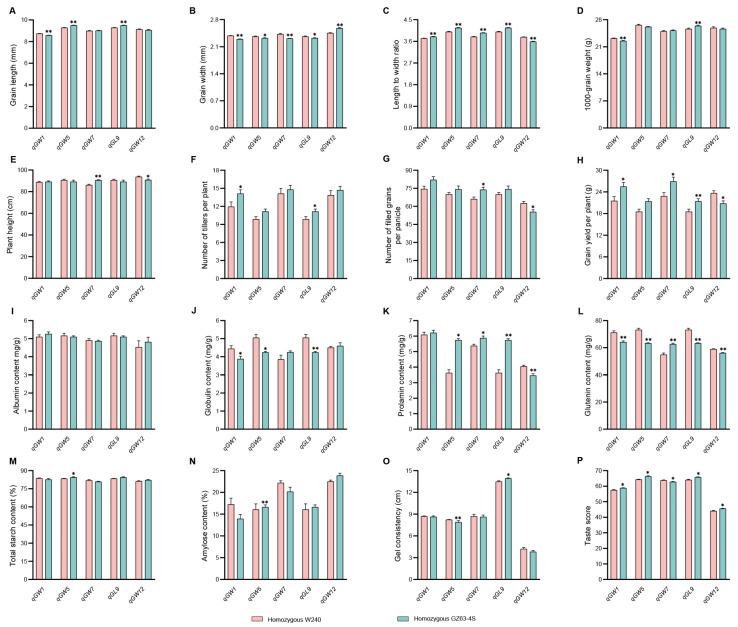
Phenotypes of rice yield and quality comparison among *qGW1*, *qGW5*, *qGW7*, *qGL9*, and *qGW12* identified in this study. (**A**–**H**) Grain length, grain width, length to width ratio, 1000-grain weight, plant height, number of tillers per plant, number of filled grains per panicle, and grain yield per plant in five NILs. (*n* = 12). (**I**–**P**) Albumin content, globulin content, prolamin content, glutenin content, total starch content, amylose content, gel consistency, and taste score. (*n* = 6). All phenotypic data in (**A**–**P**) were measured from paddy-grown NIL plants grown under normal cultivation conditions. Data are represented as mean ± s.e.m. The student’s *t*-test was used to produce *p* values (*, ** indicate significance at *p* < 0.05 and *p* < 0.01, respectively).

**Figure 9 ijms-25-04149-f009:**
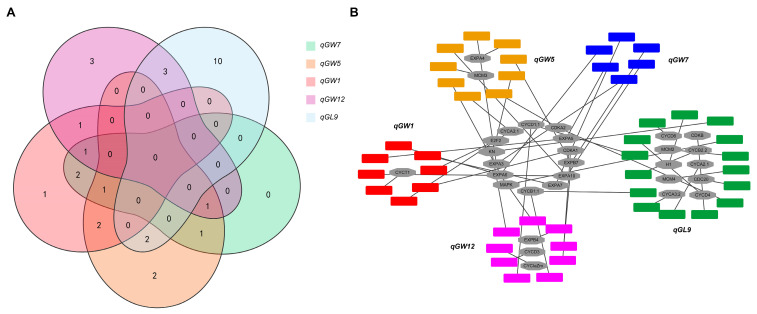
The differential expression analysis of (**A**,**B**) cell number and size genes.

**Table 1 ijms-25-04149-t001:** QTLs for grain size in the BC_1_F_2_ and BC_1_F_2:3_ populations.

QTL	Chromosome	Interval		2016			2017			Known Genes
				LOD	Add	PVE (%)	LOD	Add	PVE (%)	
*qGL1*	1	RM220	RM490	15.78	0.20	14.82	16.42	0.21	21.56	*OsCKX1*
*qGL2.1*	2	RM211	R2479	4.95	−0.05	0.42	7.98	−0.09	3.09	
*qGL2.2*	2	R2746	R2100				2.60	−0.21	14.04	*GW2*
*qGL2.3*	2	R2100	R2178				2.54	−0.09	5.84	
*qGL2.4*	2	R2198	RM262				3.62	0.07	8.92	
*qGL5*	5	RM169	RM163	4.77	−0.07	1.50	3.28	−0.06	1.47	
*qGL8.1*	8	R8354	R8464	2.73	−0.05	0.24	4.00	−0.06	1.81	
*qGL8.2*	8	R8173	R8274	2.85	−0.03	0.06				
*qGL9*	9	R9178	RM160	5.66	−0.13	11.87	9.07	−0.15	12.26	
*qGL11*	11	RM21	R1126	4.74	−0.09	2.15				
*qGL12*	12	R1224	R1226	4.46	−0.06	6.04				
*qGW1*	1	RM128	RM319				2.63	0.20	13.24	
*qGW4*	4	R4306	R4314	4.82	−0.04	5.71				
*qGW5*	5	RM122	R5147	2.68	0.10	23.32	5.24	0.04	18.33	
*qGW7*	7	RM234	RM248	4.15	0.04	6.27	2.68	0.07	15.01	
*qGW12*	12	R1224	R1226				6.29	−0.05	8.95	
*qLWR1.1*	1	RM220	RM490	3.67	0.05	3.31	3.37	0.04	2.72	*OsCKX1*
*qLWR1.2*	1	RM128	RM319				2.81	−0.03	1.71	
*qLWR5*	5	RM122	R5147	3.23	−0.05	4.37	2.52	−0.04	3.10	
*qLWR8*	8	R8173	R8274	2.78	−0.02	0.16				
*qLWR9*	9	R9178	RM160				5.93	−0.04	2.99	
*qLWR12*	12	R1224	R1226				4.02	0.04	8.81	
*qGT1*	1	RM319	R1386				3.58	−0.01	7.67	
*qGT2*	2	R2470	RM211	7.20	−0.03	21.23				

*qGL*, QTL for grain length; *qGW*, QTL for grain width; *qLWR*, QTL for length to width ration; *qGT*, QTL for grain thickness. LOD, logarithm of odds. Add, additive effect of QTL. Positive value and negative value of additive effects indicated the W240 and GZ63-4S alleles, respectively. PVE, phenotypic variance explained by the QTL.

## Data Availability

For materials, please contact the corresponding author’s email address.

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
