# Peer review of "Fine Mapping of Five Grain Size QTLs Which Affect Grain Yield and Quality in Rice"

_ijms, 2024, doi:10.3390/ijms25084149_

Round 1

Reviewer 1 Report

Comments and Suggestions for Authors

     This is a well-written and informative study that identifies and characterizes several QTLs associated with grain size and weight in rice. The functional analysis of the major QTLs and their potential application in breeding are valuable contributions. This research endeavor demonstrates commendable scholarly rigor and is well-aligned with the thematic scope of the Journal. Nonetheless, in light of the comprehensive evaluation conducted, the following constructive critiques and recommendations are tendered for the authors' contemplation and incorporation, aimed at fortifying the scholarly merit of the manuscript:

Ø  The abstract provides a comprehensive overview of the study, summarizing the objectives, methods, results, and implications effectively. However, there are a few areas that could be improved for clarity and precision-

·       Redundancy must be avoided: The abstract mentions the importance of grain size for yield and quality twice (lines 3-5 and 30-32). The last sentence of the abstract (line 22) reiterates information already mentioned earlier (line 21).

·       Missing information: The abstract mentions five major QTLs being validated and fine-mapped (line 13), but it doesn't specify which traits these QTLs are associated with.

·       Inaccuracy: The sentence "Grain weight exhibits a strong correlation with grain yield" (line 30) is inaccurate. While grain weight can contribute to yield, it is not the sole factor, and other factors like number of grains per panicle also play a significant role.

Ø  In the introduction, while the overview of the importance of grain size regulation in rice production is well articulated, there seems to be an overreliance on listing genes and pathways involved without sufficient contextualization. It would be beneficial to briefly explain the significance of each pathway or gene mentioned in the context of grain size regulation before delving into the specific genes and pathways.

Ø  The citation information needs to be completed as indicated in the authors instruction section. Additionally, the format of citation needs to be consistent throughout the document.

Ø  There are minor typographical errors such as inconsistent formatting (e.g., inconsistent spacing before and after punctuation marks) that should be corrected for uniformity.

Ø  In the methods section, it would be helpful to provide more detailed explanations of the techniques used for QTL analysis, SEM analysis, and expression analysis. This would enhance the reproducibility of the study.

Ø  The discussion of the results is thorough and insightful, particularly regarding the identification and characterization of major QTLs. However, there could be more emphasis on discussing the novelty and significance of the findings in the context of existing literature on grain size regulation in rice.

Overall, the manuscript presents valuable insights into the genetic basis of grain size regulation in rice and its implications for breeding high-yielding, high-quality varieties. With some minor revisions and clarifications, it has the potential to make a significant contribution to the field.

Comments on the Quality of English Language

A thorough check on the language is required

Author Response

Kindly refer to the attachment.

Reviewer 2 Report

Comments and Suggestions for Authors

The reason for this decision is:

This manuscript does not fulfill the standards established for the journal to be considered for publication.

First, a major error is made in mapping seed size. It is QTL mapping using BC1F2:3 of BC1F2, which is not fixed. In Figure 1, although the length/width ratio is an essential value for classifying rice seeds, it was not investigated. In Figure 1, in order to use a thousand grain weight, you must perform a correlation analysis using three factors to use the thousand grain weight data. Due to these errors in materials and survey items, the manuscript contains fundamental errors that cannot be corrected through author revision.

Author Response

Kindly refer to the attachment.

Reviewer 3 Report

Comments and Suggestions for Authors

The manuscript presents an interseting topic about investigating five novel QTLs for grain yield characters, which are important as selection criteria in rice breeding program for grain yield improvement. However, I have several notes for the manuscript in order to improve the manuscript quality.

1.     Lines 95-97: Authors state both GL and GW exhibited strong positive correlations with GT and TGW.  From Table S2, the coefficient correlation among them range only for 0.105 to 0.350 for GT, which indicates a weak correlation, and 0.216 to 0.659 for TGW that indicate weak to moderate correlation.

2.     Lines 107-108: Why was qGL1 not choosen as a QTL that will be analyzed further? This QTL has higher LOD and PVE that the other selected QTL.  What is the reason for choosing the five QTLs instead of choosing qGL1?

3.     Authors did not mention what is the density of genetic map used in the QTL and Fine mapping analyses.  Please mention it.

4.     There is no clear correlation data between grain yield characters (grain widht, grain length) with grian quality.  Please include it either in the manuscript or in the supplementray files.

5.     There in no explanation in the method section about the DEG analysis between five QTLs and Cell number and size genes

6.     There is no discussion about the relationship of the gene expression analysis and the QTL contribution to the grain yield characers. Please add the discusison about it.

7.     There in no explanation about the method/procedure/criteria of selecting inviduals from BC1F2:3 population that will be developed into specific QTL NIL.  Please explain it in the manuscript.

Comments on the Quality of English Language

There is no significant correction, except for several typos. 

Author Response

Kindly refer to the attachment.
